# Teledermatology for Common Inflammatory Skin Conditions: The Medicine of the Future?

**DOI:** 10.3390/life13041037

**Published:** 2023-04-18

**Authors:** Fabrizio Martora, Gabriella Fabbrocini, Matteo Megna, Massimiliano Scalvenzi, Teresa Battista, Alessia Villani, Luca Potestio

**Affiliations:** Section of Dermatology, Department of Clinical Medicine and Surgery, University of Naples Federico II, Via Pansini 5, 80131 Napoli, Italy

**Keywords:** telemedicine, acne, hidradenitis suppurativa, psoriasis, atopic dermatitis, COVID-19, COVID-19 vaccine, teledermatology, inflammatory disorders, skin disease

## Abstract

Background: The COVID-19 pandemic period revolutionized daily clinical practice. Several strategies were adopted by clinicians to avoid reducing treatment for diseases without the risk of spreading the infection. Among the adopted strategies, telemedicine played a key role. In this scenario, several tools were used, including e-mails, phone calls, video calls, support groups, and messages. Fortunately, the COVID-19 pandemic period seems to be at an end. However, the use of teledermatology appears to be an excellent strategy for the future as well. Indeed, several patients may benefit from teledermatology. Objective: In this manuscript, we aim to investigate the use of telemedicine in the dermatological field to point out how this tool may become the mainstay of future medicine. Only the use of teledermatology with common inflammatory skin conditions have been reported herein. Materials and Methods: Investigated manuscripts included metanalyses, reviews, letters to the editor, real-life studies, case series, and reports. Manuscripts were identified, screened, and extracted for relevant data following the PRISMA (preferred reporting items for systematic reviews and meta-analyses) guidelines. Results: A total of 121 records were identified from the analyzed databases. However, only 110 articles were assessed for eligibility. Finally, 92 articles were selected at the end of the literature research for our review. Conclusions: Teledermatology should be considered as a viable option for the dermatologist for the future. We believe that the pandemic has strengthened this service, and this will allow for ever better development in the future. Guidelines regarding the use of teledermatology are required as well as additional improvements for the future.

## 1. Introduction

Telemedicine has been defined by the World Health Organization (WHO) as the following:

Delivery of health care services, where distance is a critical factor, by all health care professionals using information and communication technologies for the exchange of valid information for diagnosis, treatment, and prevention of disease and injuries, research, and evaluation, and for the continuing education of health care providers, all in the interests of advancing the health of individuals and their communities [1].

Dermatology, radiology, and pathology are perhaps the most visual specialties in medicine, making them ideally suited for modern telemedicine techniques, as has been shown in several recent studies investigating the feasibility and reliability of this useful tool [2,3]. Telemedicine has generally demonstrated high levels of concordance in diagnosis and management plans when compared with face-to-face consultations [2,3].

Telemedicine was already introduced in clinical practice several years ago in the field of dermatology [2]. However, its use was limited to a small number of visits, and many doubts, coupled with a lack of experience on the parts of doctors and patients, made teledermatology an impractical tool. Recently, the SARS-CoV-2 (COVID-19) pandemic revolutionized daily routines. Indeed, the introduction of COVID-19 government restrictions and the “stay-at-home” policy changed daily lifestyles. Similarly, clinical practice completely changed, requiring the use of new tools to enable therapeutic continuity while reducing the risks of getting the infection. Among these new tools, telemedicine played a crucial role in the fight against the pandemic. Indeed, the COVID-19 pandemic has made it possible to “reuse” this service [4,5], implementing and improving some aspects to counter the spread of the virus worldwide. 

In particular, in the field of dermatology, telemedicine has played a key role for several reasons: to reassure patients about the continuation of biologic treatments [6], to continue to assist patients telematically through the platforms made available to physicians, to provide explanations to patients regarding the development of possible dermatologic manifestations during or post COVID-19 infection [7,8,9], and to continue to assist patients with chronic inflammatory diseases such as psoriasis, atopic dermatitis, and hidradenitis suppurativa [10,11,12,13,14,15].

The purpose of this review is to analyze how telemedicine has been a viable alternative for various dermatological diseases against COVID-19 infection and to demonstrate that teledermatology can be considered the medicine of the future.

## 2. Materials and Methods

For the current review, literature research was performed using the PubMed, EBSCO, Embase, Google Scholar, Cochrane Skin, and MEDLINE databases (until 20 January 2023). Research was performed by using and matching the following terms: “SARS-CoV-2”, “COVID-19”, “pandemic”, “telemedicine”, “teledermatology”, “telemedicine”, “skin manifestations”, “cutaneous disease”, “acne”, “psoriasis”, “atopic dermatitis”, “hidradenitis suppurativa”, “eczema”, “atopic eczema”, “treatments”, “biologics”, “biological drugs”, and “inflammatory skin diseases”. Investigated manuscripts included metanalyses, reviews, letters to the editor, real-life studies, case series, and reports. Manuscripts were identified, screened, and extracted for relevant data, following the PRISMA (preferred reporting items for systematic reviews and meta-analyses) guidelines. The performed research with the PRISMA checklist has been reported in Figure 1. The bibliography was also analyzed to include articles that could have been missed. Only English language manuscripts were included in our work. Manuscripts regarding the use of telemedicine in other medical fields were excluded, as well as manuscripts investigating the use of teledermatology for the management of other cutaneous diseases (e.g., rosacea, infestations, etc.) and studies investigating the use of this tool for the management of skin cancers such as basal cell carcinoma, squamous cell carcinoma, and cutaneous melanoma.

The dermatologic conditions considered for our review were acne, psoriasis, atopic dermatitis, and hidradenitis suppurativa. These conditions were selected because more studies were conducted for these conditions.

This article is based on previously conducted studies and does not contain any research with human participants or animals.

The extracted data of interest contained baseline demographics (age, gender, quality of life, and severity), the number of participants, treatment regimens, and outcome evaluation. Our primary outcome was DLQI.

The quality of the enrolled studies was appraised by two investigators (F.M. and L.P.). Through evaluating methodology, the overall risk of bias was judged as low, high, or some concern. If discrepant opinions existed, the third author (A.V.) was consulting for arbitration. 

## 3. Results

A total of 121 records were identified from the analyzed databases (PubMed, Ovid, Scopus, Embase, MEDLINE, Google Scholar, and Cochrane Library databases). However, only 110 articles were assessed for eligibility, since duplicate manuscripts and ineligible articles were excluded. Finally, 92 articles were selected at the end of the literature research for our review, excluding manuscripts that did not meet our inclusion and exclusion criteria. In particular, we focused our attention on four chronic inflammatory cutaneous disorders that during the pandemic could not be abandoned by dermatologists precisely because of the chronic nature of the diseases, as they required continuous follow-up visits to assess disease severity and to allow for therapeutic continuity [16,17,18]. Moreover, these pathologies were also evaluated due to their high prevalence in the general population. The diseases researched were acne, psoriasis, atopic dermatitis, and hidradenitis suppurativa. 

### 3.1. Psoriasis

Psoriasis is a chronic inflammatory cutaneous disease affecting up to 3% of the worldwide population that requires frequent follow-ups and treatments that are ongoing to avoid relapse of the disease. In particular, frequent attention is necessary for the management of moderate to severe forms of psoriatic disease, where patients take systemic treatments with immunosuppressive drugs or biologic therapy [19,20,21,22,23].

Because of this, the role of telemedicine for these patients became critical during the pandemic period to enable them to continue their usual therapies without interruptions and to monitor possible adverse events that may be related to the treatment [24,25,26].

We selected several papers in the literature where telemedicine was used for the management of psoriasis during the COVID-19 pandemic.

First of all, Andersen et al. [27], through a retrospective review of the teledermatology database in the Faroe Islands, which included a group of 18 islands in the North Atlantic, concluded that teledermatology is necessary in specific settings, particularly in rural areas, whereas in non-rural areas it must be selective for conditions that can benefit from this service. Beer, Chambers, Pearlman, and Frühauf conducted studies where it was confirmed that online interactions were both well accepted by patients but more importantly preferred over in-person visits [28,29,30,31].

In contrast, Yi et al. [32] confirmed in their study another very important fact, which is that telemedicine services have been infrequently used by elderly patients, as these patients often have great difficulty with digitization; this is a very important point, as there is a need to make this service accessible and easy to use for elderly patients as well in the future.

Similarly, in the literature, there are various works where the clinicians themselves have confirmed their preference for telemedicine services [33,34,35]; the authors conclude that this service has guaranteed the reduction of visits and consequently reduced the spread of the COVID-19 infection but has also permitted the healthcare system to save time and money.

Gisondi et al. [36] conducted one of the largest studies on telemedicine presented in the literature, with a total of 246 patients treated with biologics. In this study, 48% of patients (*n* = 118) preferred telemedicine to in-person visits. This resulted in a reduction in the risk of contracting the COVID-19 infection.

Another study with a large number of patients was conducted by Tinio et al. [37]; this study involved 424 psoriasis patients who made consultations via phone calls. A high level of patient satisfaction with this methodology was found.

Studies have also been conducted to evaluate the impact of telemedicine on mental health and depression in patients with psoriasis. Young et al. conducted a 12-month randomized study where no statistically significant differences were found compared to the past. Their study was a 12-month randomized controlled equivalency study evaluating the impact of teledermatology on the mental health and depression of psoriasis patients, and they found no statistically significant differences compared to in-person visits.

All these previous studies allow us clinicians to reflect on the use of teleservices in psoriasis management. Indeed, telemedicine, or rather teledermatology, has proved to be a reliable, effective tool to offer patients continuous therapeutic management, avoiding treatment discontinuation or interruption. While there are some points that certainly need to be improved for the future, it was essential during the pandemic to notabandon this type of patient, and perhaps we can take advantage of what has been a necessity and still offer patients this type of consultation service in the future.

### 3.2. Acne and Hidradenitis Suppurativa

Acne and hidradenitis suppurativa are two chronic inflammatory skin conditions with a high prevalence among the general population and with a high impact on patients’ quality of life [38]. 

As regards acne, the use of teledermatology for its treatment was proven to be crucial to the management of this condition during the era of the COVID-19 pandemic. Patients suffering from acne, in fact, need personalized therapies and continuous follow-up [39,40].

Currently, there are several studies in the available literature that evaluated the effectiveness and safety of this service for acne patients even before the COVID-19 pandemic [41,42].

The studies conducted on acne include a large number of patients, which shows how important it was to refine this service to avoid abandoning numerous patients.

In a retrospective analysis by Kazi et al. [43] that examined 951 synchronous and 1672 asynchronous acne visits, the authors reported that the preference for synchronous teledermatology over asynchronous for complex medical dermatology was statistically significant (*p* < 0.05).

There has been a debate regarding the prescription of isotretinoin. In the literature, there is evidence of studies such as the one by Gu et al. [44], who studied the types of visits for a total of 480 patients and concluded that topical drugs were prescribed more than systemic drugs, also concluding that care must be taken when prescribing isotretinoin in this manner.

Moreno Ramirez et al. [45] conducted a prospective study, where they concluded that the prescription of isotretinoin is feasible with teledermatology.

Lee et al. [46] analyzed 1233 virtual visits. This study showed that most elderly patients preferred the phone-only mode of the visit over the video mode preferred by younger adult patients.

Villani et al. [47] conducted a long-term study involving 213 patients; the authors concluded that 50% of the patients preferred telematic services over face-to-face visits.

This issue is very important today, as different methods of teledermatology services have been proposed, including video calls, phone calls, WhatsApp and Facebook support groups, image evaluation, and emails. It must be considered that not all patients are positive towards digitization. Therefore, in conclusion, we believe that the pandemic has allowed us to refine this service to make it more accessible to all patients.

Compared with the other dermatological conditions mentioned above, hidradenitis suppurativa deserves a different discussion.

There were fewer studies conducted on this condition for several reasons: first, the condition involves areas such as the groin, axillary or breast area, and scrotal region [48]; patients were certainly not in favor of showing such intimate areas on digital platforms in these cases.

To date, to the net of our knowledge, the following studies are in the literature: Patel et al. [49] conducted a study comparing two patient groups of about 40 people undergoing both video consultations and face-to-face visits. Most of the patients concluded that they preferred the in-person mode so as not to show their intimate areas on video.

Kang et al. [50] conducted an interview on the Facebook platform, where they asked 355 patients about their preference, and again most patients chose the mode of in-person visits over telemedicine services. Ruggiero et al. [51] conducted an interview with 54 patients, and 22.5% of patients said they preferred in-person visits. The authors concluded that the service needs to be improved in the future to provide more privacy for patients.

In conclusion, although the results look very promising for other chronic inflammatory diseases, regarding hidradenitis suppurativa and teledermatology there is much to be done in terms of safety and service effectiveness. Hidradenitis suppurativa is a very difficult disease to treat, and patient management becomes crucial. There have been several studies in the literature where different authors have employed different strategies with good results to manage hidradenitis suppurativa patients undergoing biologic therapy or for patients who were left behind during paradoxical reactions during the pandemic period [52,53,54,55,56,57,58].

### 3.3. Atopic Dermatitis

Atopic dermatitis, also known as atopic eczema, is a chronic inflammatory skin condition characterized by inflammation, redness, and irritation of the skin, with intense itching [59,60]. It affects both children (up to 30%) and adults (up to 10%) [59], with a high impact on patients’ quality of life. Globally, the pandemic period initially caused several inconveniences to these patients who require frequent specialist visits and continuous monitoring. especially for patients treated with biotechnological drugs such as dupilumab, tralokinumab, or other drugs [61,62,63,64,65,66]. The use of telemedicine has been fundamental in this period to ensure continuity of care for this type of patient [67]. The European Academy of Allergy and Atopic Dermatitis Clinical Immunology has proposed that telehealth could be useful for disease severity monitoring (using validated tools to assess the severity of the disease in a mobile app), therapeutic education, patient communication, medication reminders, and research [68,69]. A recently published retrospective study showed that the accuracy of atopic dermatitis diagnosis through telemedicine service was 84.4% [70]. In total, however, 72% of patients were managed in telemedicine, while the remaining 28% preferred face-to-face visits with their dermatologist [70].

Ragamine et al. [71] conducted a questionnaire-based study investigating the impact of COVID-19 on clinical and psychological symptoms and satisfaction with care. A total of 913 consultations (with 466 individual children) were conducted during the first COVID-19 wave in 2020, while 698 (with 391 individual children) and 591 consultations (with 356 individual children) were conducted in 2019 and 2018, respectively [71]. The results show how the use of the telemedicine service increased in the second round compared to 2018–2019 [71].

There are other studies in the literature that have evaluated the impact that telemedicine has had on atopic patients; all these data confirm how this service has been safe and effective and has consequently guaranteed limiting the spread of COVID-19 [72,73,74,75].

In conclusion, the problem of privacy remains. Atopic patients are often children, and it is difficult to convince parents to show their images on video. For this reason, the service must be implemented with encrypted services that allow for total privacy of the transmission of images. Moreover, the genital area is often involved in patients with atopic dermatitis, increasing a patient’s hesitancy to have a consultation using telemedicine.

## 4. Discussion

At the beginning of our review, we cited the definition of telemedicine given by the World Health Organization (WHO) [1]. We believe that this definition succeeds in best defining the goals of telemedicine [1]. The use of this service has been fundamental during the COVID-19 pandemic. The application of this technology by all healthcare professionals has been commendable, as this has allowed for communication and information sharing with thousands of patients; has brought advantages in term of prevention, treatments, and diagnosis; and has been a reason for continuous training for all health professionals.

Certainly, this service, by virtue of the COVID-19 pandemic, has been expanded both in terms of services offered (emails, video calls and phone calls, applications for smartphones and tablets, support groups, short message services) and in terms of remote support provided to patients so as not to abandon them. Our review showed the main strengths of telemedicine in terms of safety and effectiveness but also in terms of satisfaction for both patients and physicians. Indeed, teledermatology demonstrated itself to be a valuable weapon in reducing the number of face-to-face consultations and the spread of the COVID-19 infection during the pandemic. However, there have also been reports in the literature where the telemedicine service has instead shown shortcomings. A study conducted by dermatologists highlights both the advantages and disadvantages of this service [76]; specifically, among the disadvantages, the most frequent reports were low image quality, inadequate viewing of the skin during video consultations, and several technical problems. Bhargava et al. [77] reported teledermatology as one of the most important factors influencing the burn-out of dermatologists during the pandemic. There are also studies that have shown that another limitation of teledermatology has been misdiagnoses. Often, lesions analyzed through this service have led to misdiagnoses precisely because of the quality of the images received [78,79]. Another disadvantage of this service arises, or may arise, from the inability of many patients to share and photograph their lesions correctly, resulting in difficulties in the differential diagnosis process. There are several reports in the literature where this differential diagnosis problem is highlighted [80,81,82].

Moreover, commonly used clinical scores (e.g., Psoriasis Area Severity Index, Eczema Area Severity Index, Body Surface Area, Hidradenitis Suppurativa Severity Index, etc.) cannot be used in telemedicine, requiring new effective scores to assess disease severity by using clinical pictures. On the contrary, patients’ quality of life (Dermatology Quality of Life Index, etc.) as well as scores analyzing the level of pruritus (e.g., Pruritus—Numerical Rating Score, etc.) should be used during a visit conducted via teledermatology. Therefore, we cannot completely assess the severity of the disease by telemedicine currently, but we can evaluate the impact of the dermatological condition on daily living.

In summary, despite the fact that several advantages have been demonstrated by the use of telemedicine (e.g., reduction of face-to-face visits, allowance for continuity of care in remote areas, patients’ and clinicians’ satisfaction, cost-effectiveness, reduction of waiting lists, etc.), several limitations remain (e.g., accessibility, need for internet connection and technological devices, standardized scores, etc.). Surely, the increased use of this tool will reduce the limitations, making teledermatology an indispensable tool during daily clinical practice. However, with the end of the COVID-19 pandemic period, the use of teledermatology is likely to decline. Thus, new challenges are coming to define the correct use of telemedicine in daily routines.

In conclusion, we believe that the main role of telemedicine has been not only to be able to assist dermatologic patients with chronic inflammatory diseases undergoing therapy of any nature, but also to play a key role in recognizing adverse reactions of a dermatologic nature during COVID-19 infection or post COVID-19 vaccination. There have been many reports related to these two events of new onset of dermatologic disease or worsening of pre-existing disease [83,84,85,86,87,88,89,90,91,92,93,94,95,96,97,98,99,100,101,102,103]; this service has provided confidence and security to patients who without dermatologic consultation would have discontinued key therapies or who would, in our view, not have completed the vaccination course, thus causing an increase in COVID-19 cases worldwide. Nowadays, despite the end of the pandemic period, telemedicine seems to be an integral part of clinical practice. Certainly, in the future, the use of this service must be established with rules and recommendations shared by international guidelines.

The issue of privacy remains a problem to be solved for the future; a medico-legal solution should be pursued, such as electronic signed consent from patients who accept this service. Finally, we need platforms that offer excellent services such as image quality and that are easily accessible to reduce the barriers for the elderly.

## 5. Strengths and Limitations

The main strength of our study is the use of PRISMA guidelines to review current literature. The main limitations are the exclusion of articles regarding the use of telemedicine in other medical fields as well as the exclusion of manuscripts investigating the use of teledermatology for the management of other cutaneous diseases (e.g., rosacea, infestations, etc.) in addition to acne, hidradenitis suppurativa, psoriasis, and atopic dermatitis.

## 6. Conclusions

There were several measures applied during the COVID-19 pandemic period to prevent the exponential spread of the virus. Certainly, teledermatology was crucial; in our review, we emphasized both the effectiveness and safety of this service but also pointed out flaws that we hope will be improved in the future. Certainly, our review has highlighted the benefits that this service has offered during the pandemic period. In particular, we wanted to underline how it has been possible to guarantee a good assistance service for patients suffering from chronic dermatological pathologies such as psoriasis, atopic dermatitis, acne, and hidradenitis suppurativa; the treatments in progress were not interrupted, and, when necessary, it was possible to guarantee a visit through the platforms available, resulting in a favorable opinion from the patients themselves.

Finally, in this review, we have also highlighted the problems of this service. In particular, elderly patients have had difficulties with connections to the various platforms available, and unfortunately privacy has not been respected. These factors have led to difficulties for these patients. For the future, it should be emphasized that this service will certainly have a favorable opinion from patients if an excellent video call service is guaranteed rather than a telephone call; we also believe that the published works have certainly highlighted how this service is more useful for follow-up visits where patients are already undergoing treatment in comparison to a first dermatological visit where diagnostic errors could be made.

Work will still have to be done in the future to solve these problems and to, on one hand, guarantee full privacy for patients and, on the other hand, make accessibility to these services easier, even for patients who are not very familiar with telematic services.

We may conclude by saying that the outcome for patients in general has certainly been very positive and that telemedicine can certainly be considered the medicine of the future in the coming years, especially if the limitations we have highlighted in our review are overcome.

## 7. Future Direction

As of today, teledermatology should be considered as a viable option for dermatologists for the future. Recently, the COVID-19 pandemic period led to the improvement of teleservices, increasing their use in daily clinical practice. With the end of the COVID-19 pandemic period, we believe that teledermatology will remain an integral part of medicine, thanks to its high profile in terms of effectiveness and safety that has been previously showed. Certainly, more studies are needed to further evaluate its possible role, also considering the limitations highlighted in our manuscript. Finally, guidelines regarding the use of teledermatology are required as well as further improvements for the future.

## Figures and Tables

**Figure 1 life-13-01037-f001:**
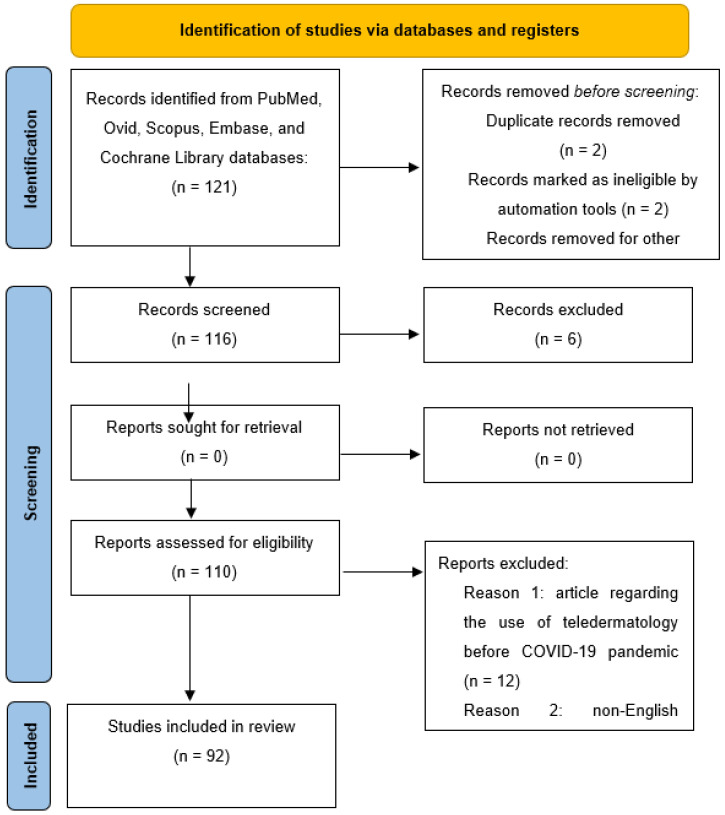
PRISMA Checklist.

## Data Availability

Data are reported in the current study.

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
