# Peer review of "Teledermatology for Common Inflammatory Skin Conditions: The Medicine of the Future?"

_life, 2023, doi:10.3390/life13041037_

Round 1
Reviewer 1 Report
Teledermatology is an important topic and it is obvious that it has a role to play even in non-pandemic time. This topic has been already extensively published in the literature.
It is interesting how the authors divided this review into four different inflammatory conditions, rather than incorporating all the other important aspects related to teledermatology.
The biggest issues I have with this paper are:
1) The omission of skin cancer assessment in this review. Dermatology is largely divided into "rash" and "lesion". The assessment of skin lesion is very different from a rash. There are also known issues with telederm due to the inability to palpate lesion and perform dermoscopic examination. I feel that cutting out essentially half of dermatology needs to be explicitly stated in the title as well as in the abstract of this article. You may have to rename it e.g. Teledermatology for common inflammatory skin conditions etc.
2) There isn't any nuanced discussion about how teledermatology is employed e.g. store-and-forward or live teledermatology consultation, differences between telehealth over the phone or over the video call. How the resolution of images or video has an impact on ability to make assessment. Photos/videos may not allow assessment of whole body and other sites.
3) Differences between initial consultation vs follow up visits. Telehealth works well for follow up of patients who generally are doing well e.g. monitoring of systemic medications. It may be less effective than face-to-face in making diagnosis of a complex skin condition.
4) The English language used here is slightly clumsy. This paper would benefit from Professional English editing service to polish its readability and also fix typographical errors e.g. line 16, 78
Author Response
Dear Reviewer,
Thanks a lot for your time and for your corrections.
1) We listened to your advice and changed the title. The choice not to include tumors is because there are fewer works in the literature, we believe that telemedicine has been more useful for chronic inflammatory diseases because diagnosing or following a tumor in telemedicine is very difficult
2) We welcomed your comments with great interest and added these considerations in the part of the conclusions.
3) We welcomed your comments with great interest and added these considerations in the part of the conclusions.
4) We have proofread all the text with a native speaker and corrected typographical errors as you suggested.
Reviewer 2 Report
Well written paper. Easy to read and useful collection material on the importance of teledermatology.
Perhaps the only comment is to soften the phrase: "Dermatology is perhaps the most visual specialty of medicine..."
In my opinion, radiology and pathology should be included in the sentence. Also visual specialties without the need to have the patient in a face-to-face consultation.
Author Response
Dear Reviewer,
Thanks a lot for your time and your comments
We are honored and happy that you enjoyed the work.
We followed your advice and included radiology and pathology in the sentence as you requested.
Thanks a lot
Best regards
Reviewer 3 Report
Dear Authors,
From my point of view you have done a very interested review. I believe it is original, well done and correctly written. Teledermatology is in progress and during the pandemic it was promotes, thus dermatology improve our knoweldge on it.
Author Response
Dear reviewer,
Thanks a lot for your time and for your comments.
We are honored that you enjoyed the work.
We fully agree with what you say about the importance of this service in the pre and post pandemic era.
Best regards
Reviewer 4 Report
The present work tries to show evidence in the literature that supports the use of teledermatology as the methodology of the future.
I think initially that the title is very ambitious since it only addresses the study of three dermopathies and in all of them detects problems for their widespread and widespread use. Therefore, the title should be somewhat more modest and stick to the objective of real that is to show evidence of a possible use of teledermatology in some skin conditions, nothing more.
The methodology needs a review since it does not mention the PRISMA protocol or reference it and is the place to do so. The Flow chart (which is not a check list, please correct this term) needs correction. There is information cut inside the boxes and there are leftover boxes with values 0, please remove.
Throughout the text authors are named, but not the year of publication as prescribed by the Vancouver standard. Please include the year in all cites.
In the results of the review I miss a table for each section that visually explains what has been reviewed and what are the advantages and disadvantages of using teledermatology for each disease. It should include the number of patients who support this technique versus those who are not supporters.
There are some small points in the text and they need to be corrected:
-line 25. A point is missing
-Line 41. I miss a reference
-line 60-62. avoid question. Make a sentence, for example: this work wants to demostrate that teledermatology could be considered the medicine of the future ….or similar
-line 69. biologic need a “
-line 70. diseases!. Correct it
-line 233 and 236. Telehealth is not the same that telemedicine????, please explain it. If it is the same, please change it.
-lines 256 to 261. Repeat definition it is not necessary. Please remove it.
Conclusions, like the title, are too ambitious and should conform to what has been reviewed and found. It is not advisable to include opinions not supported by objective data.
It would be very appropriate to name the necessary improvements so that teledermatology could be an option for the future if they are made. It is necessary to specify which points are not possible today, such as the digital difficulty for the elderly, data protection, technical limitations of users and other aspects addressed in the review.
I encourage the authors to modified the manuscript according the suggestions in order to take it into account before its acceptance as a possible publication.
Author Response
Dear reviewer,
Thanks a lot for your time and for your comments.
Here our responses point by point
1) We have changed the title as requested to specify that we have selected common chronic inflammatory diseases. In our opinion, the question in the title makes it less ambitious but at the same time poses a fair reflection. We hope you enjoy the new title.
2) In the text in the methodology section the prism method was widely explained, we also report it here and we have underlined it in red in the text so that you can check.
Manuscripts were identified, screened, and extracted for relevant data, following the PRISMA (preferred reporting items for systematic reviews and meta-analyses) guidelines. The performed research with the PRISMA checklist has been reported in Figure 1.Instead, as requested, we checked the table and eliminated the errors.
The value you see as 0 is because this is how the research was and like all prism guidelines it should be reported.
We referred to the new PRISMA guidelines where that research field is reported.
3) All citations have been taken directly from Pubmed or Scopus and have been carefully checked one by one. Probably when there is no year there are two reasons: either it is still being published or the journal does not foresee it, therefore we have inserted it.
4) Regarding the request for the tables, we are sorry but we don't think it is very useful.
All the studies in the literature do not mainly deal with what you asked for. There is no many studies comparing who uses this service and who doesn't and not always in all studies. The tables you requested would therefore not be complete with all the data, while in the text we have analyzed the favorable and unfavorable points of the cited authors study by study.
5) Thanks for reading carefully, we've edited any mistakes you told us and deleted the sentence as you requested
6)We have already explained this comment extensively in the conclusions in the old text, we have underlined it in the new one, hoping that in your opinion it could be the right way.
It should be emphasized for the future that this service will certainly have a favorable opinion from patients if an excellent video call service is guaranteed rather than a telephone call; we also believe that the published works have certainly highlighted how this service is very useful for follow-up visits where patients are already undergoing treatment compared to a first dermatological visit where diagnostic errors could be made.
Work will still have to be done in the future to solve these problems and guarantee, on the one hand, full privacy for patients and, on the other hand, make accessibility to these services easier, even for patients who are not very familiar with telematic services.
We may conclude by saying that the outcome for patients in general has certainly been very positive and that telemedicine can certainly be considered the medicine of the future in the coming years especially if the limits we have highlighted in our review are exceeded.
Round 2
Reviewer 1 Report
There is no nuanced discussion about the various methodologies in teledermatology itself e.g. store and forward or live telemedicine.
Reviewer 4 Report
Although the authors have made an effort to better explain the methodology, I still think that the results could be greatly improved with a table showing the relevant findings, but it is not required for article acceptance. It is a matter for the readers and to facilitate the understanding of the results.